# Economic and Social Impact of Huanglongbing on the Mexico Citrus Industry: A Review and Future Perspectives

Hernán Villar-Luna [ID], María Elena Santos-Cervantes, Edgar Antonio Rodríguez-Negrete [ID], Jesús Méndez-Lozano [ID] and Norma Elena Leyva-López *[ID]

Departamento de Biotecnología Agrícola, Centro Interdisciplinario de Investigación para el Desarrollo Integral Regional (CIIDIR), Unidad Sinaloa, Instituto Politécnico Nacional, Guasave 81101, Sinaloa, Mexico; hernan_villarl@yahoo.com (H.V.-L.); msantos@ipn.mx (M.E.S.-C.); erodriguezn@ipn.mx (E.A.R.-N.); jmendezl@ipn.mx (J.M.-L.)
* Correspondence: neleyval@ipn.mx

**Abstract:** The citrus industry is of great importance in Mexico, with an estimated production value of USD 2.4 billion and the potential to generate up to 28 million employees per year. Approximately 69,000 Mexican families depend on this sector. However, it is under serious threat from the disease known as Huanglongbing (HLB). The rapid spread of this disease has caused significant economic losses, impacted the livelihoods of citrus growers, and led to substantial job losses. Currently, HLB is not under control in Mexico, and as the management strategies applied have proven to be ineffective, yields have not been recovered to pre-HLB levels, and production costs have significantly increased. Therefore, it remains the most destructive citrus disease. This review paper describes the current situation of the disease, its economic and social impact, and the measures adopted for its management in Mexico. Future perspectives highlight advances in research based on new biotechnological tools as an eco-friendly management alternative. Practical suggestions to address HLB are also mentioned in our analysis.

**Keywords:** *Candidatus* Liberibacter asiaticus; citrus greening; pest management; economic importance

## 1. Introduction

Citrus fruits stand out globally for their high economic value, with major production occurring in countries such as China, Brazil, India, Mexico, and Spain [1]. These fruits are destined for either fresh consumption or processing. In America, Mexico is one of the leading countries in the citrus market, specifically for sweet oranges (*Citrus sinensis* L. Osbeck), limes (*C. aurantifolia* [Christm] Swingle and *C. latifolia* Tanaka), and lemons (*C. limon* L. Burm.f.) [2]. Mexico boasts a cultivated area of 678 thousand hectares, a total production of 9.3 million tons valued at USD 2.4 billion, with Veracruz, Tamaulipas, and Michoacán being the leading producing states; therefore, citrus farming significantly contributes to the economy of the country [1,3]. However, the citrus industry worldwide faces several challenges that potentially impact its yields, such as those caused by various phytosanitary factors: Huanglongbing (*Candidatus liberibacter* spp.) and its vector the Asian Citrus Psyllid (*Diaphorina citri*); Citrus Canker (*Xanthomonas citri* subsp. citri); Citrus Variegated Chlorosis (*Xylella fastidiosa*); Gummosis (*Phytophthora* spp.); Citrus Black Spot (*Phyllosticta citricarpa*); Post-Bloom Fruit Drop (*Colletotrichum acutatum*); Citrus Tristeza Virus (CTV) and its vector Brown Citrus Aphid (*Toxoptera citricida*); Citrus Leprosis Virus (CiLV) and the mites that spread it (*Brevipalpus* spp.). Some of these diseases are present in Mexico, except for the Citrus Black Spot and CiLV. In 2023, the Ministry of Agriculture, Livestock, Rural Development, Fisheries, and Food (SAGARPA acronym in Spanish), through the National Agriculture and Food Health, Safety and Quality Service (SENASICA acronym in Spanish), implemented the Citrus Pest Control Campaign to carry out prevention and control actions against the aforementioned pests. The management of these diseases is carried out mainly

through chemical control targeting the vector insects, as appropriate, and, to a lesser extent, biological control and cultural practices [2,4,5].

Among these diseases, the most significant is Huanglongbing (HLB) or citrus greening, categorized as the most destructive for this crop [6,7]. HLB is caused by a fastidious bacterium limited to the phloem of the genus *Candidatus* Liberibacter, with three confirmed associated species: *Candidatus* Liberibacter asiaticus (*C*Las), *Candidatus* Liberibacter africanus (*C*Laf), and *Candidatus* Liberibacter americanus (*C*Lam). Due to their uncultivable nature in artificial media, studies on these bacteria are limited [7,8]. *C*Las is the most destructive, persistent, and globally distributed species, while *C*Laf is only present in Africa, and *C*Lam was identified in Brazil. These phytopathogens are primarily spread by vector insects, with *C*Las and *C*Lam transmitted by the Asian citrus psyllid (*Diaphorina citri*) and *C*Laf by the African psyllid (*Trioza erytreae*). These insect vectors are rapidly spreading to HLB-free citrus-producing areas [9]. The development of HLB symptoms is the consequence of molecular, cellular, and physiological changes, and may also be associated with host defense response. HLB symptoms include yellow shoots, blotchy, yellowing, asymmetrical mottling, corky mid-ribs of leaves, and upright small leaves (Figure 1). Fruit symptoms include reduced fruit size and premature fruit drop, hugely decreasing fruit quality and production; fruits are misshaped, lopsided, and when cut open, show discolored columella. Orange juice extracted from the HLB-affected fruits has shown to be off-color and off-flavor due to the high accumulation of volatiles in response to the disease. As the disease progresses, the tree loses productivity, and the commercial quality of the fruits is drastically reduced due to the low photosynthetic efficiency and phloem obstruction that inhibits the translocation of photoassimilates [2,10]. Yield loss is mainly due to the premature abortion of fruits, which can reach rates of 30% to 100%, depending on the disease severity [11]. These symptoms can be confused with nutrient deficiencies (zinc or manganese) or those of other phloem-limited pathogens (Spiroplasma and Phytoplasma). Similar symptoms to those of the Citrus Tristeza Virus (CTV) can also be observed in fruits [12,13]. The presence of HLB in Mexico was detected in 2009 in the state of Yucatán in backyard Mexican lime trees. However, the vector *D. citri* has been present in the country since 2002 in the states of Campeche and Quintana Roo [14,15]. By early 2015, the disease was present in 23 of the 24 citrus-producing states in Mexico, creating a scenario with a high impact that could result in the loss of over 19 million jobs across the country [16,17]. To contribute to the understanding of HLB, its economic and social impact, and its environmental effects, this review presents the most significant contributions from the science community in Mexico.

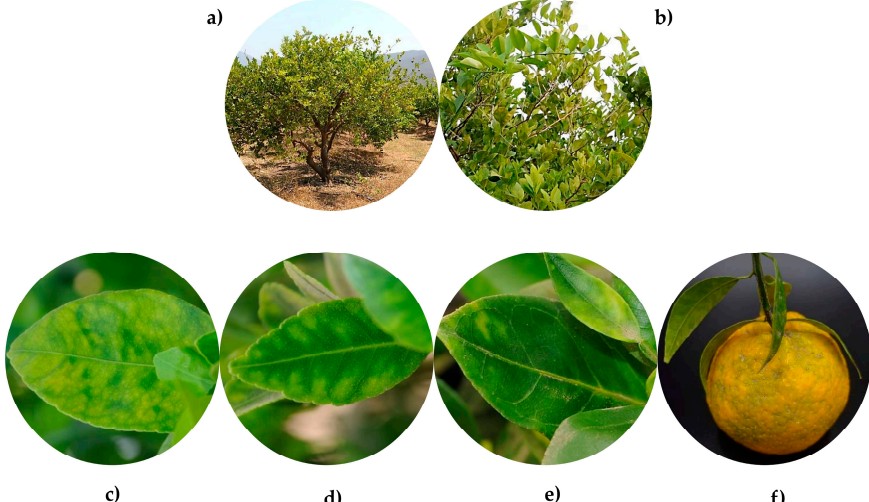

**Figure 1.** Symptoms of Huanglongbing disease in citrus. (**a**) Tree canopy greening; (**b**) Tree defoliation and dieback; (**c**) Asymmetrical mottle and yellowing in leaves; (**d**) Blotchy-mottle leaves; (**e**) Corky veins; (**f**) Small and blotchy fruits.

## 2. Epidemiology and Current Status of Huanglongbing in Mexico

Huanglongbing (HLB) arrived in the American continent in the 2000s, with the first outbreaks of the disease being detected in two of the most important citrus regions on this continent. Initially, in 2004, it was identified in the state of Sao Paulo, Brazil, and in 2005, in the state of Florida, USA [18,19]. By 2006, trees with HLB symptoms were reported in Havana, Cuba [20]. Due to the proximity to Florida and Cuba, in Mexico, in 2008, the SAGARPA, through the SENASICA, implemented phytosanitary measures to mitigate the risk of introduction and spread of the *C*Las bacteria on Mexican territory. This included a national phytosanitary campaign against HLB in all citrus-producing states of the country. Additionally, the vector *D. citri* has been present in the country since 2002, and widely dispersed in all citrus zones [15,21]. The initial actions were preventive, based on the experience of other countries in the Americas, where orange trees were the most affected by HLB. For this reason, orange orchards in Mexico were considered to be at higher risk. The phytosanitary activities carried out between 2008 and 2009 included orchard exploration for symptom detection and sampling of plants and psyllids for diagnosis. Additionally, citrus crops up to four years old localized next to bodies of water or abandoned were established as sentinel orchards (sites for epidemiological surveillance) [21]. Despite timely actions, HLB detection was achieved in the summer of 2009 in the state of Yucatán, infecting backyard Mexican lime trees. Given the severity of the symptoms shown by the trees, it is estimated that the pathogen could have established approximately two years earlier. By the end of that year, the disease was also present in the states of Quintana Roo, Nayarit, and Jalisco, with 12 municipalities in these four states already reporting the presence of HLB (Figure 2) [11,14,22–24]. Following these detections, the Emergency Mexican Official Standard NOM-EM-047-FITO-2009 establishing phytosanitary measures to mitigate the risk of the introduction and spread of citrus Huanglongbing was published in 2009 [25]. By February 2010, the presence of the disease increased from 12 to 15 municipalities, and monitoring the disease allowed for new outbreaks to be identified in the states of Campeche, Colima, Sinaloa, and Michoacán [24,26,27]. Given these events, the goal of the campaigns shifted to maintaining the phytosanitary status under HLB protection through timely disease detection and the implementation of phytosanitary actions to prevent its spread; therefore, to the aforementioned activities, others were added, such as removal of positively diagnosed plants (citrus and *Murraya paniculata*) in commercial orchards, urban areas, and nursery stock, and chemical control of the psyllid vector; in addition, training and diffusion for growers were carried out, and plant production in certified nurseries covered with anti-aphid nets was mandatory [21,25]. Even with the implemented phytosanitary measures, three years after the first detection, HLB had spread to 13 citrus-producing states of Mexico [28]. Subsequently, the epidemic exponentially expanded to 21 states in 2015, reaching 24 citrus-producing states by 2018 (Figure 2) [29,30].

The rapid spread of the vector is attributed to natural phenomena such as tropical storms and hurricanes, since the movement of *D. citri* alone is not significant [24]. During 2009–2011, prolonged droughts occurred in the country due to the El Niño phenomenon, which increased the levels of *D. citri* adults, favoring the proliferation and establishment of *C*Las in Mexico. Due to the environmental conditions, citrus crops suffered severe water stress, which triggered the epidemic in the Pacific region [31]. The most significant impacts of the disease occurred in the states of Colima, Michoacán, Jalisco, and Nayarit; events manifested during the year 2011, categorizing this region as high-risk, with incidence levels of symptomatic trees reaching up to 90% [21,27,32]. Over time, the HLB dispersion scenarios behaved differently from the two initial outbreaks in the Yucatán Peninsula and the Pacific regions; a key point to consider is that different factors can be crucial in the adaptation of the disease, such as management, host type, and climatic conditions [11,33,34]. Although the first detection of HLB disease in Mexico was in the Yucatán Peninsula, it showed a slow dispersal process; while in the Pacific region, it spread faster. An explanation for this phenomenon is that the Pacific region has a high-density commercial production of mostly sour citrus trees, such as the Mexican lime, which is more susceptible to HLB in

this region. In contrast, sweet oranges predominate in the Yucatán Peninsula and usually grow on backyard trees. These events caused the priorities of the HLB campaign to change, shifting from the orange crop to the Mexican lime crop [21,27,33,35].

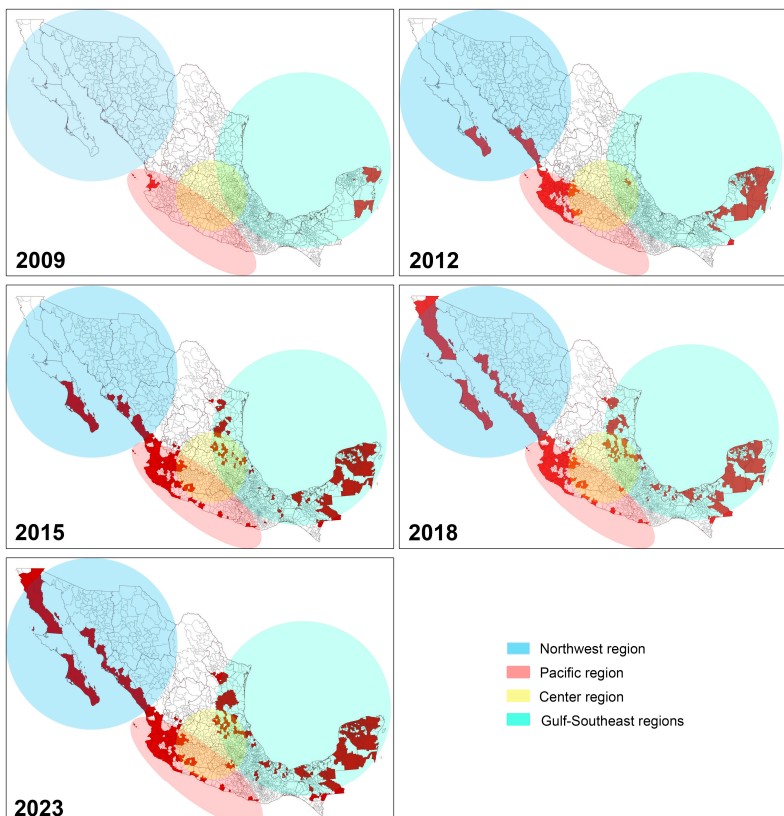

**Figure 2.** Distribution of Huanglongbing (HLB) in Mexico from 2009 to 2023. The map was performed using MapChart (https://www.mapchart.net accessed on 27 April 2024) with the annual information reported by the National Agriculture and Food Health, Safety and Quality Service (SENASICA, acronym in Spanish) [4,24,28–30]. The color of the bubbles indicates the productive region in Mexico.

Epidemiological studies on the spatio-temporal dispersion of HLB in Mexico are limited, and the insufficient application of these approaches can reduce the chances of success in disease management. A study conducted in the Yucatán Peninsula on HLB dispersion gradients, based on vector monitoring, incidence, and severity, from the first disease hotspots in Yucatán and Quintana Roo, revealed an east-to-west dispersion pattern favored by wind. The most extended HLB spread distances ranged from 25 to 82.6 km in 12 months [36]. The disease dispersion pattern closely resembles reports from Brazil, with a spread distance of 45.9 km per year, and in Florida, USA, an absolute dispersion rate of 19.5 km per year was reported [27,32,37,38]. In Mexico, two epidemiological scenarios for HLB were categorized based on disease occurrence and intensity: high intensity (Pacific Region) and low intensity (Yucatán Peninsula). Management strategies can be implemented based on epidemic intensity, considering host susceptibility and inoculum load [32]. Inoculum load, affecting the epidemic speed and distance of dispersion, requires different approaches for primary inoculum through eradication and secondary inoculum through vector control [32,39]. The HLB epidemic behavior in Mexico is diverse. Recent epidemiological studies based on the *C*Las inoculum load in trees and vectors suggest variable epidemic rates depending on the productive region. The Pacific region remains a highly chronic region, the Gulf southeast region is considered moderate, and the center and northwest regions are considered to be moderate–low [40]. In addition to differences in epidemiological intensity, a recent analysis of *C*Las genomic diversity, based on next-generation sequencing supported by machine learning analysis, revealed two clearly

defined clusters: Geno-group 1 with strains from central and northern Mexican states and one from California, USA, and Geno-group 2 with strains from southern Mexican states, the Yucatán Peninsula, and one from Florida, USA. It was also detected that these groups carry two different types of prophages. The findings support the hypothesis that the introduction of HLB into Mexico may have occurred through two distinct events. Geno-group 2 seems to still be restricted in the Yucatán Peninsula; this suggests that the measures taken by the Mexican government have generated a positive effect. On the other hand, strains of Geno-group 1 spread rapidly from west to east toward central Mexico [41]. Currently, disease monitoring continues; however, since 2019, the strategy has been modified to a "Campaign against Citrus Regulated Pests", focused on HLB, Citrus Tristeza Virus (CTV), and monitoring of Citrus Leprosis Virus (CiLV), due to the recent outbreaks of the last two years [39]. Lately, up to the latest annual report published by SENASICA, the current situation of HLB in Mexico is reported to be present in 41% of the total hectares allocated for citrus cultivation, spanning 351 municipalities in 24 citrus-producing states (Figure 2) [4].

## 3. Economic, Social, and Employment Impact Caused by Huanglongbing

The production of citrus fruits in Mexico represents a significant activity for the national economy, with 678 thousand hectares allocated for this purpose and an annual production of 9.3 million tons (Figure 3) [3]. Mexico ranks fourth globally in citrus production, with 88% designated for domestic consumption and 12% for exportation [1,42]. Oranges (53%) and limes (34%) are the most important contributors to the total production, with grapefruits, mandarins, and tangerines constituting the rest, estimated at a value of USD 2.8 billion [3,43]. Additionally, Mexican citrus farming is an important economic support for approximately 69,000 Mexican families and has the potential to generate up to 28 million workdays annually. This impact is particularly notable in the states of Michoacán, Veracruz, Oaxaca, Colima, Tamaulipas, Jalisco, Tabasco, Guerrero, Yucatán, and San Luis Potosí [42,44,45]. Employment opportunities within this agro-industry span daily laborers and field professionals, as well as packaging, industrialization, transportation, and trade. Plant growers (nurseries) and input suppliers also benefit from citrus cultivation [46]. From the 1980s until 2013, limes and oranges were the primary job generators in national fruit cultivation, ranking among the top 10 crops regarding employment demand. Due to the social and economic importance of citrus, the Mexican government considers citrus farming a high-priority activity [42,47].

However, this cultivation faces numerous challenges threatening production, with diseases caused by viruses and bacteria being the main culprits. These diseases significantly alter the landscape of citrus farming in Mexico. The most significant changes began in 2009 when Huanglongbing (HLB) was detected in the country [48]. Alarming scenarios were forecasted for the primary sector, with losses of over 19 million jobs in the face of high impact of HLB nationwide; these effects were expected to be primarily in the orange crops due to their larger cultivated area [14]. However, the reality was different, as the Mexican lime cultivation turned out to be the most affected [3]. A particular characteristic of this disease is its latency period of 6 to 12 months, during which infected trees are asymptomatic and act as a source of inoculum. This feature makes management strategies less efficient, increasing production costs [12,27]. This disease forces growers to adapt to the new dynamics of citrus farming to confront it, and the strategies include using certified citrus plants, eliminating symptomatic trees, and improving management practices (irrigation, fertilization, timely pest control, etc.) to maintain stable yields. Additionally, the productive life of the infected trees may be reduced to 5 to 8 years [12,48]. These effects result in economic losses, reduced production volume, reduced yields due to the removal of diseased trees, and reduced cultivated area, causing significant changes in the socioeconomic aspects that impact the quality of life for growers and workers. An example of this in Mexico is the case of the Mexican lime wherein, up to the present date, the cultivated area of this crop has not been able to recover [3]. It is documented that

the Mexican lime has been the most affected, especially in the state of Colima, where the disease has spread exponentially, affecting tree vigor, and reducing yield potential.

The Mexican lime production experienced a decline of over 105 thousand tons. In terms of cultivated area, from the 28 thousand hectares in 2008, approximately 10 thousand hectares were lost since the arrival of HLB in Colima by 2020 (Figure 4) [3]. It is important to note that, as the profitability of Mexican lime declined in terms of the cost–benefit ratio, some citrus growers considered agricultural reconversion as an alternative. This involved modifying the production pattern by incorporating alternative crops with greater agronomic, social, and economic viability. However, this strategy raises concerns for regions with a long tradition of citrus cultivation, negatively affecting the economy, particularly for small-scale growers who lack the means to switch crops [49,50]. In 2010–2014, approximately 5000 hectares of citrus were replaced in Colima. This shift included an increase in the cultivation area of some perennial crops like sugarcane, banana, and papaya, and the introduction of others like pineapple [17]. There has been a slight recovery of about 2 thousand hectares, culminating in the year 2022 with an estimated 21 thousand hectares (Figure 4) [3]. The impacts on production are reflected in symptomatic Mexican lime trees that produce fruits smaller in size, with a reduction of up to 45.8% per square meter of canopy. The yield per tree decreased from 57 kg to 23 kg in trees with over 75% of severity. With a severity of 50%, the yield does not exceed 30 kg/tree. The state of Colima recorded an average yield of 17.8 t/ha in 2010, but with the arrival of HLB, the yields plummeted to 9.3 t/ha, and currently stand at 14 t/ha [3,51]. To recover economic profitability, the decision was made to coexist with the disease by using integrated management, improved irrigation, and fertilization. The applied technological packages increased production costs from USD 1528/ha to USD 1792/ha but maintaining tree yield, making the cultivation profitable again. It is noteworthy that, before the arrival of HLB in the state of Colima, a total of 402,125 tons of Mexican lime were produced, decreasing to 166,805 tons at its most critical production point. Since 2015, there has been a recovery, with production reaching 297,117 tons in 2022. These imbalances resulted in an increase in production value, ranging from USD 97 million in 2010 to USD 153 million in 2022, partly due to a decrease in product supply and an increase in demand (Figure 4). It is important to note that the elevated product costs can impact the economy of the final consumer [3,52]. HLB has become endemic in the Mexican lime and Persian lime-producing areas in the states of Colima, Michoacán, Jalisco, and Nayarit, causing a significant economic and social impact. In Colima alone, an estimated 2435 full-time jobs in field labor and harvesting were lost, equivalent to 300 annual workdays between 2010 and 2012 [46].

In other citrus varieties such as the Persian lime (*Citrus latifolia*), for example, in the state of Yucatán, fruits from symptomatic trees suffered a 17.3% reduction in weight and an 18.6% reduction in juice volume; in the state of Veracruz, the effects are comparable. These weight reductions resulted in estimated losses of 2.4 t/ha in both states, associated with severity level and *C*Las titer [16,53]. In Tamaulipas, Valencia orange fruits exhibited variations in color and size, ranging from 76.1 mm in diameter in asymptomatic trees to 65.1 mm in infected trees. The quality and physicochemical properties are also significantly affected, with higher pectinmethylesterase activity and lower juice turbidity stability in infected fruits. The sugar content (°Brix) in healthy fruits is up to 11.1, and in diseased fruits, this decreases to 8.5. These conditions potentially affect the commercial quality of the product [54]. As the disease is present in small production areas in the state of Tamaulipas, it is not monitored, and it is only documented that more than 500 trees have been eliminated, and fruits from oranges with symptoms do not meet commercialization standards and are destined for the juice industry. It is known that growers in this state invest in agricultural inputs in order to combat the HLB vector. However, these investments lead to higher production costs [55,56]. Production losses have not been accurately quantified for orange and other citrus in Mexico; thus, in some regions of the country, the degree of impact on orchards is unknown.

In Veracruz, Marrs orange trees with combined infection of CTV and HLB exhibit serious yield impacts, from 12 kg/tree in asymptomatic trees to a reduction of 2 kg/tree with advanced symptoms. It is essential to note that, in this region, crops are mostly rainfed; therefore, environmental conditions such as droughts influence yield [57]. Mexico exhibits high vulnerability to HLB in its agri-food sector, with most small-scale citrus growers (63.2%) located in municipalities with high marginalization. These farmers have limited capacity to respond to a phytosanitary contingency due to their constrained budgets and difficulty accessing information, making crop health investment difficult. In contrast, medium-scale (30.6%) and large-scale (6.2%) citrus growers have better access to information about phytosanitary measures and more significant potential to implement them, including the option of crop conversion in extreme epidemic cases [50]. Recently, during the "1st Meeting on Citrus Yellowing: Current Situation and Challenges" convened by the National Council of Humanities, Sciences, and Technologies (CONAHCyT, acronym in Spanish), alarming conditions were announced for orange cultivation in the state of Veracruz affected by a possible complex of phytopathogens converging in the region (CTV and CLas). Notably, the presence of CLas with incidences ranging from 94% to 98.7% was reported. In just one year, 90% of orchards in three municipalities were destroyed by this phenomenon, resulting in losses exceeding 50% of production for the 2022–2023 cycle [58,59].

Despite the economic and production losses caused by HLB in Mexican citrus, some analyses of competitiveness and the growth potential of this agro-industry document that the dedicated area and, consequently, production increase each year (Figure 3). The motivation behind this is the growing demand for citrus products, driven not only by their economic significance but also by the extensive nutraceutical benefits they possess [43,60]. The growth of citrus cultivation seemed constant from 1990 until the 2008–2013 period, when a setback occurred. This could be attributed to various factors, including the international financial crisis of those years, but it also coincides with the introduction of HLB (Figure 3). Despite this, Mexico remains a significant global supplier of citrus [61].

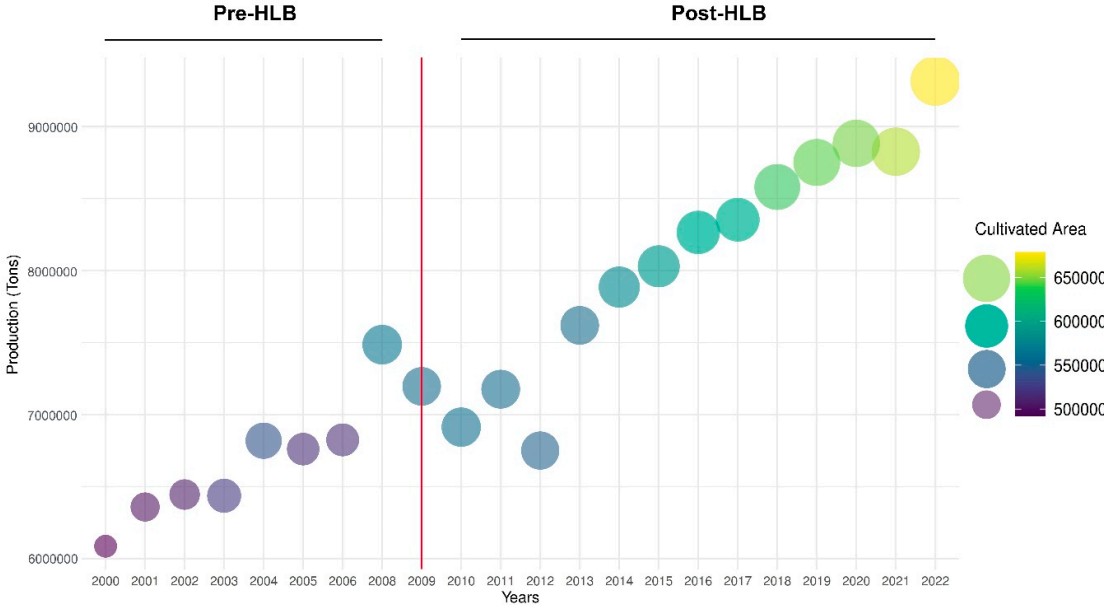

**Figure 3.** Total citrus production in Mexico (millions of tons), pre- and post-detection of Huanglongbing. The red line indicates the year of first detection, the size and color of the bubbles indicates the changes in the cultivated area during the period from 2000 to 2022. This graphic was created using R Software (v4.3.2; R Core Team) with FAO data [1,62].

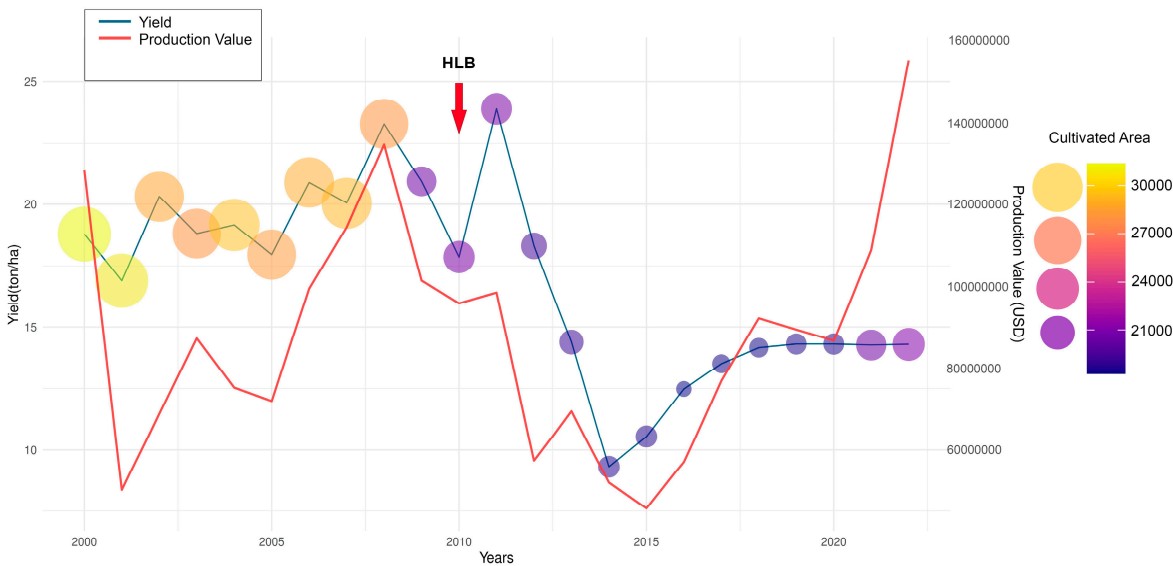

**Figure 4.** Yield, cultivated area, and production value of Mexican lime in the state of Colima from 2000 to 2022. The size of the bubbles and color indicates the changes in the cultivated area; the red arrow indicates the year of the first detection of HLB in the state of Colima [3]. This graphic was performed using R Software (v4.3.2; R Core Team) with data published by SIAP [3,62].

## 4. Control and Management Actions Implemented in Mexico

One of the first actions to mitigate the risk of the introduction and spread of HLB in the citrus-producing states of Mexico was the implementation of a phytosanitary campaign against HLB and the subsequent publication of the legal framework through the Emergency Mexican Official Standard NOM-EM-047-FITO-2009 [25]. The campaign established guidelines for HLB management [21]. Taking experiences from other countries like Brazil as a reference, the first measures taken by the National Agriculture and Food Health, Safety and Quality Service (SENASICA acronym in Spanish) were as follows: (a) designating the Mexican states where the presence of HLB was detected as areas under phytosanitary control; (b) prohibiting the movement of pathogen–host material; (c) retaining and destroying infected products (fresh fruit and propagating material) located in nurseries, crops, harvests, packaging, and boxes. Additionally, citrus growers were designated as responsible for vector control in commercial orchards and to use certified nursery plants for replanting and new plantings [17,25,48]. To date, different management strategies (chemical, biological, and cultural) have been implemented to confront the disease and mitigate significant economic losses. Since there is no successful method for managing HLB, the disease is treated preventively and targets both the insect vector and the causal agent [6]. From 2012 to 2018, SENASICA established monitoring and management zones through the HLB and *D. citri* Regional Areas for the Control program (ARCOs acronym in Spanish) [30,63]. The program proposed a model for delimiting large areas for monitoring *D. citri* and controlling infestation sources (chemical and biological) [45]. As of 2019, the strategy was changed to the "Phytosanitary Protection Campaign for Citrus Pests", and the monitoring zones were renamed as "Phytosanitary Epidemiological Management Areas" (AMEFIs acronym in Spanish) to protect the citrus crops [4,44]. To attempt the eradication of the HLB vector, the primary strategy applied was chemical control. In 2010, the Mexican government proposed a list of insecticides, according to the mode of action and toxicological groups: OC-Cyclodiene (endosulfan); Organophosphate (chlorpyrifos, dimethoate, metamidophos, monocrotophos, omethoate, malathion, acephate, fosmet); Pyrethroid II (bifenthrin, abamectin, fenpropatrin); Neonicotinoid (imidacloprid, thiamethoxam, dinotefuran, thiacloprid); Tetranoic Acid (spirotetramat, spiromesifen); Spinosins (spinetoram) [64]. Subsequent studies demonstrated that the insecticides abamectin and imidacloprid reduce nymph populations by over 90%, while in adult *D. citri*, imidacloprid and imidacloprid + betacyfluthrin achieve

mortalities of 100%. Another recommended insecticide is spiromesifen, which showed efficiency against *D. citri* nymphs of up to 88% [33,65]. However, in some citrus regions of Mexico, *D. citri* populations showed elevated levels of resistance to neonicotinoid and organophosphate insecticides [66].

On the other hand, since 2007, contact insecticides such as bifenthrin, malathion, chlorpyrifos, paraffin oil, and mineral oils have been used due to their low-cost and effective control of the vector during the first three years of use. However, starting in 2011, difficulties have arisen as farmers report low effectiveness [67,68]. Adults and fourth instar nymphs of *D. citri* have shown high resistance to malathion and chlorpyrifos. These cases have been documented in the state of Michoacán, and resistance to this group of insecticides has also been observed in other citrus-producing regions of Mexico [68]. The biological effectiveness of chlorpyrifos can be improved by combining it with paraffin oil, and this combination achieves nymph mortalities of up to 71%. On the other hand, insecticides belonging to the toxicological groups of mineral oils, S-ethyl heterocyclic organophosphates, and spinosins exert the most significant selection pressure [69]. Currently, the insecticides abamectin, cypermethrin, chlorpyrifos, dimethoate, and imidacloprid are used [70]. Recently, an alarming resistance of *D. citri* to imidacloprid was reported in the state of Veracruz, where researchers classify it as the highest worldwide [71]. To prevent the psyllid from continuing to develop resistance, an integrated management approach is recommended, involving vector monitoring and controlling infestation hotspots through insecticide applications at biologically justified times. If possible, this should be performed under a scheme that rotates toxicological groups and incorporates biological control agents. The use of biological control agents, such as natural enemies of *D. citri*, entomopathogenic microorganisms, and parasitoids, can be effective [7]. Some parasitoids are present in Mexico and have been tested against *D. citri*. For instance, nymphs are parasitized by the wasp *Tamarixia radiata*, with a parasitism rate of 60%. Other enemies of *D. citri* include *Diaphorencyrtus* sp., *Chrysoperla* sp., and ladybugs *Cycloneda sanguínea* and *O. v-nigrum*, which are found in citrus orchards [72]. Some entomopathogens, like the fungus *Hirsutella citriformis*, exhibit high virulence against adults and nymphs of *D. citri*. In bioassays, they achieved elimination rates of up to 88% and 82% for adults and nymphs of the psyllid, respectively [73,74]. Additionally, field applications have been made with strains of fungi such as *Metarhizium anisopliae*, *Cordyceps bassiana*, *Isaria fumosorosea*, and *Isaria javanica*, proven to be effective against nymphs and adults. The current campaign recommends using of *T. radiata*, *I. javanica*, and *M. anisopliae* [70,75]. When applied exogenously, oils, plant extracts, and secondary metabolites effectively work against *D. citri*. Different oils (paraffinic, cooking, citrus, and Jatropha seed), neem extracts, garlic, and onion on young lemon shoots act as repellents and reduce *D. citri* infestation. Other oils, such as anethole, verbenone, 4-ethyl-4-methyl-1-hexene, 4-allylanisole, and trans-tagetone, as well as extracts from *Foeniculum vulgare* and *Tagetes* sp., have been evaluated in vitro and proven to be toxic or repellent to adults and nymphs. The oil with the highest biocontrol potential is *Tagetes* sp., with up to 92% efficacy at a dose of 40 mg/mL [76,77].

On the other hand, Argovit™ silver nanoparticles (AgNPs) have been used against the bacteria associated with HLB (*Candidatus* Liberibacter asiaticus), applied by foliar spray and injection into the trunk of Mexican lime trees. Both application techniques achieve bacterial titer reductions between 80 and 90%. Furthermore, they decrease the starch accumulation in the phloem vessels. These AgNPs have demonstrated greater effectiveness than β-lactam antibiotics [78]. Management strategies targeted toward the host, such as cultural practices, also offer certain advantages in facing HLB disease. It is known that diseased trees exhibit nutritional deficiencies due to poor water and nutrient absorption, resulting from root system damage caused by HLB [79]. Chemical (100 N-22 P2O5-195 K2O-30 MgO), organic (compost), and combined fertilization, applied either radicular or foliar with zinc sulfate, iron, copper, manganese, and borax on Marrs orange trees, increase flowering and fruiting [57]. Likewise, black, white, and aluminum colors of plastic mulches have shown a positive effect, reducing the incidence and severity of HLB by up to

40% over 13 months. This system also decreases the number of *D. citri* adults, increasing fruit yield. Therefore, this production system is a viable alternative for citrus growers to coexist with HLB [78,80]. To maintain the profitability of the crop, integrated management approaches involving chemical, biological, and cultural practices are necessary. Different technological packages have been implemented, considering these three strategies. These actions help sustain tree productivity, and the crop becomes profitable again. In the case of the Mexican lime, since 2015, there has been a consistent recovery in production from 166,805 to 297,117 tons, with the average yield increasing from 9.3 to 14.30 tons/ha; the cultivated area has expanded from 18,633 to 20,866 ha, and the production value has risen from USD 46 million to USD 153 million by the year 2022 (Figure 3) [3,17,52].

## 5. Future Perspectives and Recommendations

HLB is threatening the citrus industry in Mexico. The rapid spread of this disease has caused significant economic losses in some regions, mainly for citrus growers. Despite this limitation, the growth of the citrus industry remains steady; however, HLB is still a threat. Citrus cultivation is a major industry in Mexico, with an estimated annual production worth USD 2.4 billion. Therefore, developing a good surveillance and early warning system is essential, as well as leveraging new computer technologies and strengthening research focused on the plant–pathogen–vector interaction. This will support the implementation of new prevention strategies and eco-friendly management. Currently, this field needs to be explored in Mexico. Research aided by novel biotechnological tools can help identify solutions to the disease and represent the leading prospects for the future. Various lines of research are currently underway in Mexico with the potential to develop efficient management alternatives. In response to infection with *C*Las, the transcriptomes of Mexican lime (*C. aurantifolia*) and Persian lime (*C. latifolia* Tan.), citrus species with a certain degree of tolerance to the disease, have been reported using the RNA-seq technique [81,82]. The transcriptomic profile of Mexican lime at two stages of the disease (asymptomatic and symptomatic) revealed an overexpression of genes associated with responses to biotic stress, such as the cell wall, secondary metabolism, transcription factors, signaling, and redox reactions. These responses are associated with HLB tolerance. Furthermore, the small RNA profile identified 46 miRNAs, including 29 known miRNAs and 17 novel miRNAs, suggesting that the regulation of *C*Las response may also be mediated by miRNAs [83,84]. The transcriptomic profiles of symptomatic and asymptomatic plants of Persian lime identified upregulated genes involved in starch and sucrose metabolism pathways, as well as genes related to biotic stress, mainly genes from plant–pathogen interaction pathways (PTI, ETI, and hormonal response). These responses can be attributed to Persian lime tolerance. Subsequently, it was documented that the tolerance of Persian lime to HLB differs from what has been reported in close citrus relatives. For example, *Poncirus trifoliata*, which is associated with the constitutive disease resistance (CDR) gene family. However, although these genes are present in the Persian lime, they do not show a significant difference between symptomatic and asymptomatic plants [82,85]. Another explored approach is an alternative management strategy targeting the vector through RNA interference (RNAi), a natural mechanism for genetic regulation and antiviral defense in eukaryotic cells. This mechanism allows for the design of double-stranded RNA (dsRNA) for use as pesticides. A method based on recombinant small interfering RNA (siRNA) generated through Escherichia coli has been evaluated against three genes (*AWD*, *SOD*, and *WNT*) crucial in the development and maturity of the psyllid. The siRNAs induced the silencing of all three genes, reducing gene expression and causing mortality in the psyllid [86]. Together, these studies can lay the groundwork and identify targets for genetic improvement and the potential of the functional RNAi system, which could be exploited to manage HLB successfully. On the other hand, attention should also be paid to developing diagnostic strategies for the timely detection of HLB, which is sometimes hindered by the high associated costs. Currently, it is carried out through the quantitative polymerase chain reaction test (qPCR). Deep learning techniques may be used to design an automated detection system that incorporates a

convolutional neural network (CNN) model to distinguish HLB from other anomalies in orange trees. In Mexico, CNNs were tested with a dataset of 953 images, which showed a 95% sensitivity to HLB. The researchers propose the development of a portable system for in-field HLB detection for small-scale growers in high marginalization regions [87].

Citrus crops have traditionally been cultivated as grafted trees on selected rootstock cultivars to improve tree performance. Although transgenic approaches have made some progress in combating HLB [88], public and regulatory concerns about transgenic plants prevent their widespread commercial acceptance. To address this, it has been proposed to explore the potential of grafting non-transgenic scions onto transgenic rootstocks. Emerging technologies like TLS-RNA fusions could facilitate approaches with transgenic rootstocks. The TLS (tRNA-like structures) technology can either be used to deliver the product from the transgenic rootstock to the non-transgenic scion tissue or combined with CRISPR/Cas9 genome editing to generate transgene-free mutants. TLS technology has the potential to fight against HLB in Citrus, and this perspective should motivate researchers to work in this field [89].

It is important to improve current production systems and consider agroecological solutions that support efficient vector control, such as intercropping and agroforestry, to increase beneficial insect populations. These alternatives can reduce dependence on agrochemicals that are likely to be overused, such as imidacloprid in orchards in the state of Veracruz. Simultaneously, the incidence of HLB in this state is alarming. A recent study reveals future projections for the years 2050 and 2070 regarding the spatial distribution of *D. citri*. There are no expected decreases in the psyllid populations; in fact, it appears in new regions where it is absent. Therefore, there is a risk of an increase in HLB. Even though these are not citrus-dedicated zones, Rutaceous species may be present, from which the vector can continue to spread the disease [90]. It is evident that significant challenges lie ahead in addressing this critical citrus disease in Mexico. Awareness of the importance of prevention and proper implementation of management measures is essential. To achieve these goals, SENASICA, the scientific community, farmers, and consumers must work together.

**Author Contributions:** Writing—original draft preparation, H.V.-L.; Conceptualization and formal analysis, M.E.S.-C., E.A.R.-N. and J.M.-L.; writing—review and editing, N.E.L.-L. All authors have read and agreed to the published version of the manuscript.

**Funding:** This research was funded by INSTITUTO POLITECNICO NACIONAL (SIP-20230970 and SIP-20240495). H.V.-L. thanks CONAHCyT.

**Data Availability Statement:** Not applicable.

**Conflicts of Interest:** The authors declare no conflicts of interest.

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
