# Peer review of "Economic and Social Impact of Huanglongbing on the Mexico Citrus Industry: A Review and Future Perspectives"

_horticulturae, doi:10.3390/horticulturae10050481_

Round 1
Reviewer 1 Report
Comments and Suggestions for Authors
General comments
The paper “ECONOMIC AND SOCIAL IMPACT OF HUANGLONGBING ON THE MEXICO CITRUS INDUSTRY: a review and future perspectives” presents an interesting overview of Huanglongbing disease on citrus in Mexico. However, this paper needs several improvements before publication. The structure of the article should be improved. For example, it would be useful to merge parts 3. and 4. (economic and sociological impacts). Part 6. should be expanded.
Particular comments
The abstract has to be re-written; it is not sufficiently informative and there are several inconsistencies. For example, line 17-18, it is indicated that: “… there is no management strategy capable of reducing its effects …” and line 19-20: “This review paper describes …. the measures adopted for its prevention and management in Mexico.” So perhaps we need to be more precise; there are disease management measures, but maybe they are not sufficiently effective to significantly reduce the impact of the disease.
The introduction is clear and well-written. Even if HLB is the most important disease for citrus in Mexico, perhaps it would be interesting to draw up a more complete review of the health status of Citrus in Mexico. It might also be interesting to present the other threats to the industry and the measures taken to avoid them; see for instance:
Barbieri, H. B., Fernandes, L. S., Pontes, J. G. D. M., Pereira, A. K., & Fill, T. P. (2023). An overview of the most threating diseases that affect worldwide citriculture: main features, diagnose, and current control strategies. Frontiers in Natural Products, 2, 1045364.
Ndo, E. G. D., Bella-Manga, F., Ndindeng, S. A., Ndoumbe-Nkeng, M., Fontem, A. D., & Cilas, C. (2010). Altitude, tree species and soil type are the main factors influencing the severity of Phaeoramularia leaf and fruit spot disease of citrus in the humid zones of Cameroon. European journal of plant pathology, 128, 385-397.
The part “Epidemiology and Current Status of Huanglongbing in Mexico” seems well done, but a few clarifications could be made:
Line 74 and line 86: It would be interesting to know more about the “phytosanitary campaign” and the “phytosanitary actions”; what exactly do they consist of?
The sentence “A study of HLB dispersal gradients in the Yucatán Peninsula, based on vector surveillance, incidence and severity over 12 months from the first disease outbreaks in Yucatán and Quintana Roo, revealed an east-west dispersal pattern favoring wind.” has to be re-written. (probably: “… favoring wind” -> “ … favored by wind”.
In the following sentence: “The most extended HLB spread distances ranged from 25 to 116 82.6 km”, is it per year?
In my opinion the parts 3. and 4. have to be merged; it seems difficult to separate economic and social impacts. In addition, certain sections (e.g. lines 161 to 175) should be moved to the introduction. (Lines 161 to 175: this part refers to physiological alterations and symptomatology).
There are also problems with the plan of the article later on: lines 285 to 289 are more concerned with control and management (not social impact).
Line 301: I suppose there is a mistake “…. in the Mexican and Persian lime producing areas in the states … “. (probably:” in the Mexican of Persian lime producing areas in the states ..”).
In part 6, it would also be interesting to consider agroecological solutions to improve vector control (such as intercropping and agroforestry, with a view to increasing populations of beneficials).
Comments on the Quality of English LanguageA careful proof-reading seems necessary; a few suggestions are made above.
Reviewer 2 Report
Comments and Suggestions for Authors
Dear authors:
This review is very good and complete, covering different areas associated with HLB. However, the abstract and the developed text have some things that could be improved and be more consistent.
1. All economic data are in Mexican pesos. They should be corrected and converted to US dollars, which are more universal.
2. When introducing a species' scientific name, it's crucial to acknowledge the expertise of those who classified it correctly. Also, the scientific name of the Mexican lime needs to be included.
3. Given the context of this review being published in a horticulture magazine, the inclusion of a figure with photos of citrus trees exhibiting symptoms associated with HLB is necessary.
4. In Figure 1, the three citrus regions described in the text should be included. Not all of us know Mexico properly.
5. The abstract needs to be more consistent with the text; it speaks of significant current losses, which are inconsistent with the increase in the area planted with citrus trees or the substantial increase in production.
6. If HLB is a significant disease, where is the damage currently? It is probably associated with an increase in production costs. The authors should consider this point more extensively.
